# Implementation of a Non-Thermal Atmospheric Pressure Plasma for Eradication of Plant Pathogens from a Surface of Economically Important Seeds

**DOI:** 10.3390/ijms22179256

**Published:** 2021-08-26

**Authors:** Agata Motyka-Pomagruk, Anna Dzimitrowicz, Jakub Orlowski, Weronika Babinska, Dominik Terefinko, Michal Rychlowski, Michal Prusinski, Pawel Pohl, Ewa Lojkowska, Piotr Jamroz, Wojciech Sledz

**Affiliations:** 1Laboratory of Plant Protection and Biotechnology, Intercollegiate Faculty of Biotechnology University of Gdansk and Medical University of Gdansk, University of Gdansk, 58 Abrahama, 80-307 Gdansk, Poland; jakuborlowski96@gmail.com (J.O.); weronika.babinska@phdstud.ug.edu.pl (W.B.); prusinski.michal98@gmail.com (M.P.); ewa.lojkowska@biotech.ug.edu.pl (E.L.); wojciech.sledz@biotech.ug.edu.pl (W.S.); 2Department of Analytical Chemistry and Chemical Metallurgy, Wroclaw University of Science and Technology, 27 Wybrzeze St. Wyspianskiego, 50-370 Wroclaw, Poland; anna.dzimitrowicz@pwr.edu.pl (A.D.); dominik.terefinko@pwr.edu.pl (D.T.); pawel.pohl@pwr.edu.pl (P.P.); piotr.jamroz@pwr.edu.pl (P.J.); 3Laboratory of Virus Molecular Biology, Intercollegiate Faculty of Biotechnology University of Gdansk and Medical University of Gdansk, University of Gdansk, 58 Abrahama, 80-307 Gdansk, Poland; michal.rychlowski@biotech.ug.edu.pl

**Keywords:** dielectric barrier discharge, plasma pencil, *Pectobacteriaceae*, soft rot erwinias, plant protection, agriculture, *Vigna radiata*, mung bean

## Abstract

Plant pathogenic bacteria cause significant economic losses in the global food production sector. To secure an adequate amount of high-quality nutrition for the growing human population, novel approaches need to be undertaken to combat plant disease-causing agents. As the currently available methods to eliminate bacterial phytopathogens are scarce, we evaluated the effectiveness and mechanism of action of a non-thermal atmospheric pressure plasma (NTAPP). It was ignited from a dielectric barrier discharge (DBD) operation in a plasma pencil, and applied for the first time for eradication of *Dickeya* and *Pectobacterium* spp., inoculated either on glass spheres or mung bean seeds. Furthermore, the impact of the DBD exposure on mung bean seeds germination and seedlings growth was estimated. The observed bacterial inactivation rates exceeded 3.07 logs. The two-minute DBD exposure stimulated by 3–4% the germination rate of mung bean seeds and by 13.4% subsequent early growth of the seedlings. On the contrary, a detrimental action of the four-minute DBD subjection on seed germination and early growth of the sprouts was noted shortly after the treatment. However, this effect was no longer observed or reduced to 9.7% after the 96 h incubation period. Due to the application of optical emission spectrometry (OES), transmission electron microscopy (TEM), and confocal laser scanning microscopy (CLSM), we found that the generated reactive oxygen and nitrogen species (RONS), i.e., N_2_, N_2_^+^, NO, OH, NH, and O, probably led to the denaturation and aggregation of DNA, proteins, and ribosomes. Furthermore, the cellular membrane disrupted, leading to an outflow of the cytoplasm from the DBD-exposed cells. This study suggests the potential applicability of NTAPPs as eco-friendly and innovative plant protection methods.

## 1. Introduction

The world population was estimated to increase to 9.6 billion people by 2050; therefore, global food production must likewise rise by 70% to satisfy the corresponding nutritional needs [1]. One of the ways to assure a sufficient amount of high-quality nourishment is by limiting the adverse effects associated with the occurrence of phytopathogens. In general, plant diseases lead to economic losses amounting to 30% of the yield if the control of pathogens and pests is not managed correctly and effectively [2]. Presently, there are over 200 species of bacterial phytopathogens that have been described [1,3]; yet the number of available control procedures remains scarce [4].

The management of bacterial diseases is regarded as challenging. Growers often decide to plant susceptible cultivars due to a lack of resistant varieties or customer preferences. This fact, together with the occurrence of environmental conditions favorable for the pathogen, results in high disease incidences [5]. For some bacterial diseases, e.g., soft rot or blackleg on potato, such resistant cultivars have not yet been marketed [4,6,7]. In comparison to fungal pathogens, a low number of chemicals intended to combat bacterial pathogens have been registered up to the present day [5]. Additionally, public acceptance of pesticides is dropping, as these substances interfere with the natural balance of the ecosystem and affect non-target organisms [8]. In terms of biological control, this method proved well under laboratory conditions; however, there were problems in the field with the establishment of large, stable-in-time populations of antagonistic microorganisms under variable weather and climatic conditions [5]. On the other hand, physical measures, mostly including hot water, steam, hot dry air, or UV irradiation [4,6], were destined primarily for storage, greenhouse, or industrial facilities. This physical, eco-friendly approach needs no registration but may impede seed germination and target beneficial microflora [6]. Notably, in the case of numerous plant pathogenic bacteria, only preventive approaches are available, i.e., planting of certified pest-free seed lots, undertaking proper hygienic cultivation procedures, or monitoring of the occurrence and dissemination of disease-causing agents [4,6].

Plant pathogenic bacteria spread in the ecosystem by diverse routes, primarily by contaminated seeds or plants, but also by soil, water, within natural vectors (such as insects or nematodes), or on the surface of non-disinfected agricultural machines [6,9,10]. A prominent contribution of the latently infected plant seeds to the dispersal of phytopathogens is associated with a reoccurrence of the previously combated disease outbreaks or an establishment of the pests in novel geographical regions [11]. In this view, the setting up of crop production from pathogen-free seeds is recommended by growers’ advisory boards, scientists, and governmental institutions throughout the world, even though it is believed to be extremely difficult to achieve [11]. In order to introduce pathogen-free seeds to the market, various preventive techniques are required, including detection of bacteria by inoculation of a susceptible host or microbiological media with the seed extract, in addition to the methods based on serology, molecular diagnostics (e.g., single, multiplex, or real-time polymerase chain reaction; PCR), or flow cytometry [4,6,11]. Despite preventive measures, various physical (heat, high-pressure, irradiation), chemical (e.g., Ca(OCl)_2_, Ca(OH)_2_, H_2_O_2_, C_2_H_5_OH, CH_3_CO_3_H, NH_3_), and biological approaches (microorganisms, bacteriophages, bacteriocins), are applied in diverse combinations for seed disinfection purposes [12]. In particular, physical treatments found application in this field, though not only the exposure time and provided temperature need to be thoroughly controlled to avoid detrimental effects on the plant material, but these approaches are also rather time- and energy-consuming, i.e., the hot water exposure needs an additional drying process, while the provision of dry air lasts approximately 5–7 days to combat certain disease-causing agents [13]. Due to the growing public demand for non-chemical, eco-friendly, effective methods of short processing times designated for seed disinfection, scientists focused on the potential applicability of non-thermal atmospheric pressure plasmas (NTAPPs). The mode of action of NTAPPs is associated with the generation of several kinds of reactive oxygen and nitrogen species (RONS), including, e.g., NO_x_, N_2_^+^, N_2_, HO•, H_2_O_2_, O•, O_2_, and O_3_ [14]. The above-listed RONS exhibit a defined redox potential that accounts for the proposed diverse applications of NTAPPs, including the ones in the agricultural sector. So far, the utilization of NTAPPs has been proposed for the eradication of several phytopathogenic microbes from cell suspensions, plant organs, contaminated fruits, vegetables, or seeds [15,16]. In terms of seeds, Selcuk et al. [17] achieved an over 3-log reduction in the *Aspergillus* spp. and *Penicillum* spp. colony-forming units (CFUs) per gram of wheat seed after 15 min of an air- or SF_6_-based low-pressure plasma treatment. These plant pathogenic fungi present on the surfaces of other seeds, including barley, oats, lentil, rye, corn, and chickpea, were also exposed to the utilized plasmas, though notably lower inactivation rates have been acquired [17]. Besides, Selcuk et al. reported no detrimental effects of the NTAPP treatment on the seed germination efficacy [17]. Regarding bacterial pathogens, Nishioka et al. [13] utilized a low-pressure plasma device, working under an Ar atmosphere, for the eradication of *Xanthomonas campestris* pv. *campestris* (currently *Xanthomonas campestris*) from the surface of *Brassica campestris* seeds. In this case, an approx. five-log reduction in the number of CFUs was noted after 10 min of plasma exposure [13]. Short periods of low-pressure plasma treatment, in contrast to the NTAPP application, did not show any detrimental effects on the germination potential of brassicaceous seeds [18]. Moreover, a low-pressure radio-frequency (RF) plasma system with a vibrating stirring device applied for two hours by Ono et al. [19] resulted in an over two-log reduction in the number of *X. c.* pv. campestris CFUs present on the surface of cabbage seeds. Other promising studies reported (i) a decrease in the disease severity of the seedling blight by 60% after planting *Burkholderia plantarii*-infected rice seeds subjected for ten minutes to the plasma irradiation [20], and (ii) a 25% increase in the resistance of tomato plants to *R. solanacearum* following a 15 s RF discharge treatment of the seeds [21].

In this study, we applied, for the first time, a NTAPP system for direct eradication of phytopathogenic bacteria belonging to *Dickeya* and *Pectobacterium* spp., family *Pectobacteriaceae* [22], primarily from spherical glass models, and subsequently from the surface of mung bean seeds. The NTAPP was generated in a dedicated, mobile, portable, and easy-to-use plasma pencil, working under a helium atmosphere and employing dielectric barrier discharge (DBD) as the NTAPP source. To the best of our knowledge, the here-chosen model plant pathogens, i.e., the members of the soft rot *Pectobacteriaceae* (SRP), have been previously subjected solely to a direct plasma action in a cell suspension [23,24] and on the surface of fresh food [25] or indirect effects of the plasma-activated liquids (PALs) [26]. In the present work, we also examined the *Vigna radiata* (mung bean) seeds germination efficacy and seedlings growth after the DBD treatment. Additionally, we used optical emission spectrometry (OES), transmission electron microscopy (TEM), and confocal laser scanning microscopy (CLSM) to decipher the antimicrobial mechanism of the DBD action towards phytopathogenic bacteria. The presented technology, after examination of its applicability under field conditions, may be a step toward incorporating plasma-based approaches into the integrated pest management strategy [3].

## 2. Results and Discussion

### 2.1. DBD-Triggered Inactivation of Bacterial Phytopathogens from Glass Spheres and Mung Bean Seeds

The analyzed plant pathogenic bacteria from the family *Pectobacteriaceae* have been efficiently eradicated from the surface of artificially inoculated model glass spheres (Figure 1). If 0.5 McFarland (McF) bacterial suspensions have been applied on glass models, the DBD-triggered percentage reductions in the amounts of colony-forming units per milliliter (CFU mL^−1^) of Ringer buffer equaled 100% for *Dickeya solani* IFB0099, 100% for *Pectobacterium atrosepticum* IFB5103, and 99.999% (5.15 log) for *Pectobacterium carotovorum* IFB5118. When bacterial suspensions of lower density, i.e., 0.05 McF, were utilized for inoculating glass beads, the resultant mean reduction rates were as follows: approx. 100% for *D. solani* IFB0099, 99.998% (4.61 log) for *P. atrosepticum* IFB5103 and 99.999% (4.93 log) for *P. carotovorum* IFB5118. In the case of 0.005 McF suspensions of plant pathogenic cells, the obtained mean inactivation rates amounted to 99.989% (3.97 log) for *D. solani* IFB0099, 99.931% (3.16 log) for *P. atrosepticum* IFB5103, and 99.975% (3.60 log) for *P. carotovorum* IFB5118. Therefore, independently of the initial level of bacterial inoculum and the investigated species, an over 3.16 log (99.931%, Figure 1) reduction rate resulted from the exposure of microorganisms present on glass spheres to DBD. From the phytopathogens tested, *P. atrosepticum* IFB5103 turned out to be the most resistant to the applied DBD treatment (Figure 1); however, the differences between diverse species (in terms of bacterial suspensions of the same optical density) showed no statistical significance (*p* < 0.05; Figure 1). It is worth underlining that the most considerable differences between repetitions were observed if the model glass spheres had been inoculated with bacterial suspensions of the lowest optical density tested (i.e., 0.005 McF; Figure 1).

As shown in Figure 2, the successful elimination of phytopathogenic bacterial cells from the surface of artificially inoculated mung bean seeds has been achieved. In terms of 0.5 McF density of bacterial suspensions, the percentage reductions in CFU mL^−1^ amounted to approx. 99.999% (5.03 log) for *D. solani* IFB0099, 99.998% (4.77 log) for *P. atrosepticum* IFB5103 and 99.999% (4.89 log) for *P. carotovorum* IFB5118. Concerning 0.05 McF bacterial suspensions inoculated on mung bean seeds, the noted mean reduction rates after the DBD treatment equaled 99.999% (5.03 log) for *D. solani* IFB0099, 99.986% (3.85 log) for *P. atrosepticum* IFB5103, and 99.998% (4.82 log) for *P. carotovorum* IFB5118. If 0.005 McF suspensions of phytopathogenic cells were used, the mean inactivation rates reached 99.914% (3.07 log) for *D. solani* IFB0099, 100% for *P. atrosepticum* IFB5103, and 100% for *P. carotovorum* IFB5118. Thus, all the recorded microbial reduction rates from mung bean seeds exceeded 3.07 log (99.914%, Figure 2), which is close to the value that we obtained in terms of the DBD-treated model glass spheres. Of the tested bacterial species, *D. solani* IFB0099 showed the highest resistance to the DBD treatment on mung bean seeds, though the noted variation (applying to bacterial suspensions of the same optical density) exhibited no statistical significance (*p* < 0.05; Figure 2). The highest differences between repeats were noted if mung bean seeds had been inoculated with *D. solani* IFB0099 suspension of the lowest investigated optical density, i.e., 0.005 McF (Figure 2).

Contrary to the work of Schnabel et al. [27], we did not observe high differences in the potency of the DBD treatment in terms of microbial eradication from the surface of glass sphere models (>3.16 log) in comparison to the studied mung bean seeds (>3.07 log). In the evoked study, a direct ten-minute DBD action reduced the amount of *Bacillus atrophaeus* spores by 2.1–2.4 logs from the utilized glass models. Meanwhile, a 0.7 log decrease in the number of the above-mentioned spores was noted on *Brassica napus* seeds [27]. This might be associated with the fact that Schnabel et al. studied bacterial endospores, not vegetative cells like in our case. The herein reported efficacy of NTAPP, being a result of the DBD treatment of plant pathogenic bacteria, is comparable to that observed by Nishioka et al. [13] after a low-pressure plasma treatment and exceeds the potency declared by Ono et al. [19], recorded after the exposure of *X. c.* pv. *campestris*-inoculated brassicaceous seeds to low-pressure air plasma with a vibrating stirring device. Although, it should be considered that in both above-mentioned studies, longer DBD-treatment periods than in our research have been applied. Referring to the work focused on the eradication of *P. carotovorum* from fresh food, it was found that 15-min exposure of artificially contaminated lettuce, carrots, apples, or strawberries to the plasma-processed air, reduced the microbial load by over 6 logs [25]. However, if shorter treatment periods (e.g., five minutes) were applied, the resulting reduction rates in the number of *P. carotovorum* cells decreased to 1–4 logs depending on the plant material used [25]. In that research also little impact on the texture, appearance, and odor of the plasma-treated apples was observed by a sensory examination performed according to DIN 10,964 standards [25]. The latter study confirmed former assumptions on the high suitability of NTAPPs for the sanitization of food products [28].

### 2.2. Impact of DBD Treatment on Germination Rate and Early Growth of Mung Bean Seedlings

Importantly, in order to retain the unaltered physiological properties of the seeds, the provided plasma treatment should not be too harsh and prolonged. Therefore, the discharge parameters of the applied NTAPP, such as the amplitude of voltage, the duty cycle, and the flow rate of the discharge gas, should have been fully optimized prior to use. Having this in mind, we focused on the DBD-generating portable device that we previously constructed and studied for medical purposes [29]. As NTAPP remains the source of an electromagnetic field, heat radiation as well as UV photons, we investigated whether the DBD exposure causes a physical damage to the plant material or alters the germination-associated properties of mung bean seeds.

The noted mean germination rates (Figure 3) of the control (0 min-), 2 min-, and 4 min-DBD-treated mung bean seeds, respectively, were as follows 84%, 87%, 58% (after 24 h); 89%, 93%, 82% (after 48 h); and 96%, 100%, 96% (after 96 h).

Two minutes of the exposure of mung bean seeds to DBD were responsible for a subtle rise by 3–4% in the germination efficacy of the NTAPP-treated seeds in comparison to the corresponding control sample (Figure 3). Though, it needs to be considered that the noted differences were not statistically significant (*t*-test at *p* < 0.05; Figure 3). Interestingly, all mung bean seeds, which had been subjected to the action of DBD for 2 min, germinated after 96 h of incubation. In terms of the 4-min NTAPP exposure, the applied dosage of DBD notably impaired (*t*-test at *p* < 0.05; Figure 3) by 26% the germination rate of the tested seeds after 24 h. The observed difference of 7% in the germination efficacy between four-minute DBD-treated *V. radiata* seeds and the control group was not statistically significant (*t*-test at *p* < 0.05) after 48 h of incubation and dispersed completely post 96 h of the plasma action (Figure 3).

The measured mean lengths of mung bean seedlings (Figure 4), grown from the control group of seeds, seeds after 2 min of DBD or seeds post 4 min of DBD, were as follows 9.7 mm, 11 mm, 6.9 mm (after 48 h), 27.8 mm, 30 mm, 25.1 mm (after 96 h).

Two minutes of DBD-based treatment of mung bean seeds caused a slight, statistically insignificant (*t*-test at *p* < 0.05; Figure 4) increase, i.e., by 13.4% or 7.9%, in the mean length of the resultant seedlings in comparison to the control group, as measured after 48 h or 96 h of incubation, respectively (Figure 4). On the contrary, four minutes of the DBD exposure of *V. radiata* seeds led to a diminution of 28.9% and 9.7% in the length of the seedlings in relation to the untreated control samples after 48 h and 96 h, respectively, from the NTAPP action. The divergences between seedlings grown from the four-minute DBD-exposed seeds in relation to the corresponding control samples turned out to be statistically significant post 48 h, as opposed to 96 h of incubation (*t*-test at *p* < 0.05; Figure 4).

The above-described results are in agreement with a former study by Schnabel et al. [27], who subjected *B. napus* seeds to the action of DBD for ten minutes. Alike the herein reported two-minute exposure of *V. radiata* seeds to DBD, a subtle increase, i.e., by 3% or 13%, in the percentage of the germinated seeds of the plasma-treated samples in comparison to the control group was observed post 24 or 48 h of incubation, respectively [27]. As in the case of the herein applied two minutes of the NTAPP treatment, the plasma dosage utilized by Schnabel et al. [27] resulted in a nearly 100% seed germination rate after 96 h. Similarly to the herein shown effects of the diverse duration of the DBD treatment, Mitra et al. [30] reported an advantageous impact on the *Cicer arietinum* seed germination rate post short (1–2 min) surface microdischarge (SMD) plasma exposures in contrast to the longer treatment periods (4–5 min). Furthermore, comparably to our study, these authors reported impeded length and dry weight of the seedlings in addition to the decreased vigor index after a prolonged plasma action [30]. Interestingly, Puligundla et al. [31] observed that the application of a corona discharge plasma jet for one minute improved the germination rate of *B. napus* seeds by 7.7% in contrast to the untreated control seeds. However, a significant reduction in the germination rate of these seeds in addition to decreases in the wet weight and total length of the resultant seedlings were noted when this discharge system was applied for over three minutes. The water content, the amount of reducing sugars or total phenolics, 2,2-diphenyl-1-picrylhydrazyl (DPPH) radical scavenging activity, color, and texture of the seedlings were not affected by the plasma exposure of the seeds. Although, some deviations in the appearance, flavor, taste, and overall acceptance of the resultant sprouts were shown if the corona discharge exposure exceeded three minutes [31]. Based on the results of Ling et al. [32], it can be hypothesized that stimulatory effects of the plasma treatment are associated with discharges generated at lower power values (60–100 W in contrast to 120 W). Besides, differences between the treated and untreated groups are more clearly visible after shorter (e.g., 24 h or 48 h) than longer post-treatment periods (e.g., 96 h, 120 h, or several days) [18,27].

### 2.3. Mechanism of the Antibacterial Action of DBD

In order to reveal the mechanism of the antibacterial activity of DBD towards plant pathogenic bacteria present on the surface of economically important seeds, first, qualitative identification of reactive oxygen species (ROS) and reactive nitrogen species (RNS) produced in the gaseous phase of the implemented NTAPP source was performed. To reach this aim, the OES analysis was carried out. As can be seen in the emission spectra of the gaseous phase of DBD (Figure 5), there was a dominance of N_2_ bands, belonging to the C^3^Π_u_-B^3^Π_g_ system, and N_2_^+^ bands, belonging to the B^2^Σ^+^_u_-X^2^Σ^+^_g_ system. Additionally, three weak NO bands of the γ-system (A^2^Σ^+^-X^2^Π) (at 226.4 nm, 236.3 nm, and 247.11 nm) and two OH bands of the A^2^Σ-X^2^Π system (at 281.1 nm and 306.4 nm) were identified. The NH bandhead at 336.0 nm was also noticed. Moreover, prominent lines of hydrogen: H_α_ at 656.28 nm, H_β_ at 486.13 nm, and oxygen (O I) at 777.2 nm, 777.4 nm, and 844.6 nm were observed. The atomic lines of the discharge gas, i.e., He I at 388.8 nm, 501.5 nm, 587.6 nm, and 706.5 nm, were also excited. Based on the performed OES analyses, it can be concluded that several RONS such as N_2_, N_2_^+^, NO, OH, NH, and O were generated in the gaseous phase of DBD.

Secondly, the TEM imaging revealed deviations in the morphology of 5-min NTAPP-treated *D. solani* IFB0099 cells in contrast to the untreated controls (Figure 6). Such abnormalities in the bacterial morphology, mostly related to changes in the smoothness of the cell surface, associated with the disruption of the bacterial cell wall, were also observed by Park et al. [33], who visualized five-minute Ar-NTAPP-treated *Staphylococcus aureus* cells in contrast to the untreated controls with scanning electron microscopy (SEM). Furthermore, Olatunde et al. [34] reported the plasma triggered deformations of *Pseudomonas aeruginosa*, *Vibrio parahaemolyticus*, *Escherichia coli*, *Listeria monocytogenes,* and *S. aureus* cells as imaged by SEM. There [34], shrinking, etching, aggregation, and harsh poration of the five-minute DBD-treated bacteria were shown. Regarding TEM-based visualizations, Miao et al. [35] revealed physical damages, including uneven distribution, condensation, and loss of cytoplasm in *E. coli* cells after the application of three or five-minute DBD.

In addition, we noted a significant elevation in the central intracellular electron density within DBD-treated *D. solani* IFB0099 cells in contrast to the untreated controls (Figure 6). Similar high electron density zones to the ones presented in Figure 6 were observed by Lee et al. [36] during studies on gram-positive (*S. aureus* and *Streptococcus mutans*) and gram-negative (*Klebsiella oxytoca* and *Klebsiella pneumoniae*) bacterial cells post-exposure to a non-thermal atmospheric pressure plasma jet on titanium discs. Wang et al. [37] attributed the presence of high electron density regions in the plasma-treated bacterial cells to the accumulated chemicals, including ROS, RNS, and acidic conditions, which caused denaturation and aggregation of DNA fragments, proteins, and ribosomes [37]. Furthermore, we herein noted electron transmission areas near the margins of DBD-treated *D. solani* IFB0099 cells (Figure 6). Analogous observations were made by Wang et al. [37], who associated the presence of these areas with an outflow of the cytoplasm from all sites of the cell due to the rupturing of the cellular membrane. Moreover, our findings were confirmed by the study of Nishioka et al. [13]. With the use of an ethidium monoazide treatment and quantitative real-time PCR, the there-reported antibacterial properties of the five-minute low-pressure plasma action have been attributed to the destruction of DNA and cellular membranes in plant pathogenic *X. campestris* cells [13]. We also discovered gaps between the cytoplasmic membrane and the bacterial cell wall in DBD-treated *D. solani* IFB0099 cells (Figure 6), which is in agreement with Feng et al. [38] who treated *E. coli* cells with silver ions. It is worth emphasizing that the antibacterial mechanism of the silver ions action is based on the generated RONS [39]. These reactive species also tend to be responsible for the antimicrobial properties of non-thermal plasmas [23].

Besides the TEM-imaged abnormal cell morphology of DBD-treated *D. solani* IFB0099 cells (Figure 6), a disruption of the bacterial cellular membrane was visualized with CLSM (Figure 7). There is a significantly higher amount of propidium iodide-stained dead or damaged *D. solani* IFB0099 cells after the DBD treatment in comparison to the untreated control samples (Figure 7). Moreover, the signal from Syto 9 stain, allowing for visualization of viable cells, is notably weaker in terms of the plasma-exposed *D. solani* IFB0099 cells in relation to the corresponding controls (Figure 7). As propidium iodide penetrates through the interrupted cellular membrane, a substantial damage to this natural cell barrier by NTAPP can be hypothesized. This explains the TEM-visualized release of the inner cytoplasmic contents (Figure 6). Analogous effects of the DBD treatment towards human bacterial pathogens were described previously [36,40,41,42,43]. For instance, the utilization of CLSM by Ziuzina et al. [42] revealed that either the direct or indirect 300 s action of DBD led to the recording of strong propidium iodide-related signals from the plasma-treated biofilms of *P. aeruginosa*. In spite of the cell disruption and the loss of bacterial viability, the thickness of *P. aeruginosa* biofilm was also significantly reduced post the DBD exposure [42]. Regarding planktonic cells, the herein presented data find confirmation in the study of Kim et al. [43], who reported a bacterial membrane interruption in *Campylobacter jejuni* based on the BacLight assay after a five-second application of DBD, generated under an oxygen atmosphere, or two minutes of DBD, generated under a nitrogen atmosphere. So far, disorganization of the bacterial membrane post plasma treatment has been attributed to the impact of plasma-generated charged particles, i.e., ions and electrons [44]. In more detail, Mendis et al. [45] suggested that the noticed cellular damage is the outcome of the accumulation of electric charge, causing enough electrostatic stress to overcome the material tensile strength.

There is a constant rise in the number of articles devoted to the use of NTAPPs for agricultural applications [16]. Even a specific term, ‘Plasma Agriculture’, has been coined for this research field [46,47,48]. A growing interest in the herein discussed novel, green, safe and efficient technology is associated with expectations that the NTAPP-based approach might fill in the gap between real and potential plant yields, which, due to the action of pathogens and pests, reaches 50–75% [47]. By now, scientists mainly focused on the utilization of plasmas for boosting seed germination and crop productivity, decontaminating wastewaters, or combating pathogens responsible for food-borne illnesses [23,46,49]. Here, a high utility of the direct NTAPP exposure, generated in a transportable and easy-to-use plasma pencil, for eradication of plant pathogens from the genera *Dickeya* and *Pectobacterium* present on plant material has been shown. The applied plasma irradiation, if it did not exceed two minutes, did not show any adverse effects on the treated mung bean seeds. Moreover, we provided the first insights into the antibacterial mechanisms of the DBD action towards plant pathogenic bacteria. The latter part of this study revealed that the contact of phytopathogenic cells with the plasma-acquired RONS, in this case, N_2_, N_2_^+^, NO, OH, NH, and O, putatively led to denaturation and aggregation of DNA fragments, proteins, and ribosomes. The bacterial cellular membrane lost its integrity, leading to the outflow of the cytoplasm from the NTAPP-treated cells. We believe that the herein described research outcomes respond to the currently suggested high demands for a close collaboration between plant pathologists and plasma chemists [47] for developing NTAPP-generating devices intended for plant tissue treatment, performing their optimization, and further testing in view of future implementation of the plasma-based technology into agricultural practice.

## 3. Materials and Methods

### 3.1. A DBD-Based Portable Plasma Pencil

DBD was used as the NTAPP source. This type of a NTAPP portable plasma pencil device was generated, developed, and optimized by our research group [29] (Figure 8). The plasma pencil consisted of a glass tube covered by the E-57 epoxy resin. In this quartz tube, two tungsten electrodes were included. A CORIAN insulator was used to provide protection against any uncontrolled electrocution. In order to assure a HV potential, a Dora Electronics Equipment supply (Wilczyce, Poland) was employed. DBD was ignited under the following discharge conditions: the frequency of the modulation: 2.17 kHz; the duty cycle: 68%; and the He flow rate: 5.0 L min^−1^. The working parameters were determined by using a digital two-channel storage oscilloscope (Tektronix, TBS 1000, Beaverton, OR, USA). He (99.999%), implemented as a discharge gas, was bought in Linde (Pszczyna, Poland). The distance between the tip of the plasma pencil and the treated materials was set to 2.0 cm or 3.0 cm by applying a digital caliper. The DBD exposure lasted from 1 to 5 min, depending on the conducted experiment as described below.

### 3.2. Bacterial Suspensions

Plant pathogenic bacteria utilized in this study (Table 1) originated from international collections of microorganisms and were stored as frozen stocks at −80 °C in 40% glycerol at Intercollegiate Faculty of Biotechnology University of Gdansk and Medical University of Gdansk (Gdansk, Poland).

The 0.5 McF suspensions of bacterial cells (Table 1) in physiological saline were prepared as described previously [26]. Briefly, the bacterial biomass was collected from the frozen stock and streaked in a reductive manner on a Trypticase soy agar (TSA) medium. Post 24 h of incubation at 28 °C, a single bacterial colony was used for inoculation of 5 mL of Trypticase soy broth (TSB). After another 24 h incubation at 28 °C with 120 rpm shaking, the overnight bacterial culture was centrifuged (10 min, 6500 rpm), and the resultant bacterial pellet was washed twice in sterile physiological saline. Then, a densitometer DEN-1B (Biosan, Riga, Latvia) was used to adjust the optical density of the bacterial suspension in physiological saline to 0.5 McF (≈1–2 × 10^8^ CFU mL^−1^ [50]). Serial dilutions of the 0.5 McF suspension of the analyzed phytopathogenic cells (Table 1) were performed for the acquisition of the corresponding 0.05 McF and 0.005 McF bacterial suspensions.

### 3.3. Antibacterial Properties of DBD towards Plant Pathogens Inoculated on the Surface of Glass Spheres

As suggested by Schnabel et al. [27], the potential utility of DBD for bacterial inactivation from plant seeds was established firstly with the use of the model glass spheres (Biospace, Poznan, Poland) of 3.5 ± 0.3 mm in diameter.

Primarily, glass spheres were immersed for 1 min in 70% ethanol. After evaporation of ethanol residuals, the models were sterilized by autoclaving (45 min, 121 °C, 1 atm). Subsequently, the glass spheres have been artificially inoculated with 0.5, 0.05, or 0.005 McF suspension of the studied plant pathogenic bacterium (Table 1), i.e., 5 spheres were incubated for 30 s with shaking in 0.5 mL of the tested bacterial suspension. After removal of the bacterial suspension residuals, 5 inoculated glass models were placed with the use of sterile tweezers in a well of a 12-well microplate (Nest Scientific, Woodbridge, NJ, USA). Such 5 spheres were treated for 1 min with the DBD plasma device. In the meantime, the 12-well plate was gently shaken to allow for a uniform DBD exposure. A 2 cm distance between the spheres and the plasma stream was kept. The control samples, i.e., glass models inoculated with the analyzed phytopathogens (Table 1) but not subjected to the DBD action, were included.

Five plasma-treated or five control glass spheres were subsequently flooded with 1 mL of a 1/4 Ringer buffer (Oxoid, Hampshire, UK) and shaken for 10 min at 600 rpm. The bacteria-containing Ringer buffer was then serially diluted to 10^−6^, and 100 µL of each dilution was plated on a TSA medium. Incubation (48 h, 28 °C) of the inoculated plates followed. The number of CFUs grown on the TSA plates was counted. The results were depicted as percentage and logarithmic reductions in the number of viable bacterial cells post-NTAPP-treatment, in comparison to the corresponding untreated control samples. The experiment was repeated three times with three technical replicates.

### 3.4. Antibacterial Properties of DBD towards Plant Pathogens Inoculated on the Surface of Mung Bean Seeds

The mung bean (*Vigna radiata*) seeds were purchased from a local market (W. Legutko, Jutrosin, Poland). At first, the seeds were immersed for 30 s in 70% ethanol. After being washed in sterile distilled water for 1 min, *V. radiata* seeds were flooded for 5 min with 3% H_2_O_2_. The disinfected seeds were washed four times in distilled water to remove any disinfectant residuals. Next, 5 mung bean seeds were submerged for 30 s in 0.5 mL of a bacterial cell suspension (Table 1; 0.5, 0.05, or 0.005 McF) and shaken. The excess bacterial suspension was discarded, and 5 seeds were transferred with sterile tweezers to one well of a 12-well microplate for a 10 min drying at 35 °C. The well, containing 5 inoculated mung bean seeds, was treated by DBD for 2 min from a 2 cm distance. Gentle shaking was assured during the procedure in order to provide uniform plasma exposure. Disinfected and inoculated with the tested phytopathogens (Table 1), *V. radiata* seeds were incorporated as the control samples.

Five DBD-treated or five control mung bean seeds were immersed for 10 min in 1 mL of 1/4 Ringer buffer (Oxoid, Hampshire, UK) and shaken at 1000 rpm. Serial dilutions to 10^−6^ of the bacteria-containing Ringer buffer were then performed. Subsequently, 100 µL of every dilution was spread on a TSA medium and subjected to 48 h incubation at 28 °C. The CFUs grown on the TSA plates were counted. The outcomes were presented as percentage and logarithmic reductions in the number of viable plant pathogenic cells in contrast to the corresponding controls. The experiment was repeated three times with three technical replicates.

### 3.5. Impact of the DBD Treatment on the Mung Bean Seed Germination Rate and the Early Growth of Seedlings

The mung bean seeds have been disinfected as described above. 5 disinfected *V. radiata* seeds have been introduced with sterile tweezers to one well of a 12-well microplate. Such well was exposed to the plasma either for 2 min or 4 min from a 2 cm distance. The microplate was meanwhile slightly shaken to ensure the homogeneous DBD treatment. Disinfected mung bean seeds not subjected to the plasma were utilized as the control samples.

Five DBD-treated or five untreated control seeds were placed on a filter paper (type MN 640 W; Macherey-Nagel, Duren, Germany) on Petri plates (96 mm diameter) and flooded with 5 mL of sterile distilled water. After having been covered with the lids, the seed-containing plates were incubated in a plant growth chamber with 16 h light (120 μmol m^−2^ s^−1^, 20 °C) and 8 h darkness (18 °C). To investigate the seed germination efficiency, the number of germinated seeds was evaluated after 24 h, 48 h, and 96 h from the beginning of incubation. In the case of the monitoring of the early growth of seedlings, the lengths of the sprouts were measured post 48 h and 96 h incubation. The experiment was repeated three times with three technical replicates; 5 seeds per replicate were included.

### 3.6. Mechanism of the Antibacterial Action of DBD

The OES was used to perform the qualitative identification of RONS produced in the gaseous phase of the studied NTAPP source. The radiation emitted by the DBD plasma was imaged on an entrance slit (10 µm) of a 0.5 m Shamrock SR-500i spectrograph (Andor, Belfast, UK). The spectrograph was equipped with a cooled to −60 °C Newton DU-920P-OE CCD camera (Andor, Belfast, UK) and two holographic diffraction gratings with 1200 grooves mm^−1^ and 1800 groves mm^−1^. The CCD camera with 1024 × 255 pixels was operated in a full vertical binding (FVB) mode, and an integration time of 0.5 s was applied. The OES measurements were acquired in the spectral range from 200 to 900 nm.

The TEM and CLSM analyses were carried out to visualize changes in the morphology of the plant pathogenic cells post DBD treatment in contrast to the untreated control samples. The 0.5 McF suspension of *D. solani* IFB0099 cells (Table 1) was prepared as described above. 0.5 mL of this suspension was exposed to the DBD treatment for 5 min from a 3 cm distance. The 0.5 McF suspension of *D. solani* IFB0099 cells that had not been exposed to the discharge was included as a control sample.

Concerning the TEM-based analysis, the staining and visualization of the samples were performed by the Electron Microscopy Section of the Faculty of Biology at the University of Gdansk as a commercial service. Briefly, the droplets of the NTAPP-treated and control bacterial suspensions were placed onto Cu-C grids CF300-Cu (EMS, Hatfield, PA, USA) with carbon films for a 1 min incubation. After removing excess liquid, the grids were incubated for 1 min in a 1% uracil acetate solution. The residues of the stain were collected using a filter paper. When the treated and control samples dried, they were visualized with a TEM instrument, model Tecnai G^2^ Spirit Bio TWIN (FEI, Hillsboro, OR, USA) at 120 kV.

Regarding CLSM, the LIVE/DEAD BacLight Bacterial Viability Kit (Invitrogen, Waltham, MA, USA) was utilized to stain DBD-treated and untreated *D. solani* IFB0099 cells according to the manufacturer’s protocol. The integrity of the cellular membranes has been assessed based on the fact that the SYTO 9 dye enters all cells, while propidium iodide diffuses solely to dead or damaged cells possessing the disrupted cellular membranes. The specimens were imaged using a CLSM Leica SP8X (Leica, Wetzlar, Germany) device with a 63× oil immersion lens.

These experiments were performed in three replicates.

### 3.7. Data Visualization and Statistical Analysis

The collected data have been visualized and statistically analyzed with R v. 3.1.3 [53]. If possible, the *agricolae* package was utilized. Levene’s and Shapiro-Wilk’s tests were applied for evaluating the equality of variances and the data normality, respectively. Depending on the fulfillment of the requirements of parametric analysis, either the Student’s or Welch’s two-sided *t*-test was implemented to assess the statistical significance of the differences between the plasma-treated and control groups. In terms of evaluating the susceptibility of different bacterial species to DBD, as the analysis of variance (ANOVA) requirements were not fulfilled, the Kruskal-Wallis test was used and followed by the post hoc applying Fisher’s least significant difference criterion with Bonferroni correction. *p* < 0.05 was utilized for all the calculations. The Figures have been assembled in Inkscape v. 0.92.3 (https://inkscape.org/).

## 4. Conclusions

In this work, for the first time, an application of a direct NTAPP treatment with DBD as a plasma source was shown effective against economically important phytopathogens from the genera *Dickeya* and *Pectobacterium*. Pathogens were artificially inoculated either on spherical glass models or mung bean seeds. All noted logarithmic reduction rates resulting from the exposure to DBD exceeded 3.07 logs (99.914%). The shorter (2 min) plasma treatment turned out to slightly promote mung bean seed germination and subsequent growth of the seedlings, in contrast to the undesirable impacts of the prolonged (4 min) DBD action. The detrimental effects of the longer DBD exposure on seed germination and seedlings’ growth were not visible or notably reduced after 96 h from the treatment. Furthermore, pioneering studies with OES, TEM, and CLSM on the mechanism of DBD action towards plant pathogenic bacteria revealed that the generated RONS, i.e., N_2_, N_2_^+^, NO, OH, NH, and O, probably caused denaturation and aggregation of DNA fragments, proteins, and ribosomes. Moreover, the outflow of the cytoplasm from the DBD-exposed cells due to rupturing of the cellular membrane was shown. In conclusion, the DBD treatment has the potential to be an innovative, efficient, and eco-friendly plant protection method and is expected to find numerous applications in the agricultural sector.

## 5. Patents

The construction of the utilized plasma pencil and application of the described NTAPP-based plant disinfection method are protected by Polish Patent Applications, no. P.429275 and no. P.438360, respectively.

## Figures and Tables

**Figure 1 ijms-22-09256-f001:**
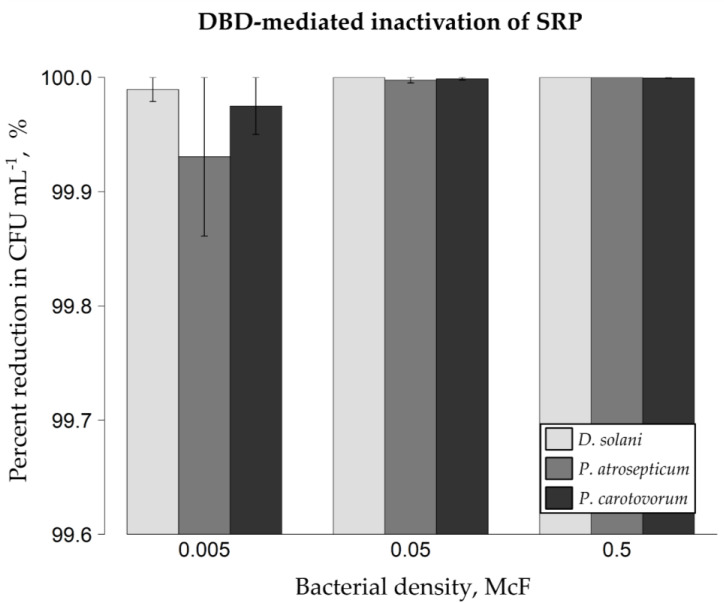
Inactivation efficacy of soft rot *Pectobacteriaceae* from glass sphere models with the use of DBD as the NTAPP source. Percentage reductions in the amount of colony-forming units (CFU) per milliliter of Ringer buffer were shown. Either 0.5, 0.05 or 0.005 McF bacterial suspension had been used for inoculating sterile model glass spheres. Means ± standard errors are depicted. The experiment was performed in triplicate with three technical replicates each. No statistically significant differences between groups were observed (Kruskal-Wallis test followed by a post-hoc analysis applying Fisher’s least significant difference criterion with Bonferroni correction at *p* < 0.05). *D. solani* IFB0099, *P. atrosepticum* IFB5103, and *P. carotovorum* IFB5118 strains were used. DBD—dielectric barrier discharge. NTAPP—non-thermal atmospheric pressure plasma. McF—McFarland scale. SRP—Soft rot *Pectobacteriaceae*.

**Figure 2 ijms-22-09256-f002:**
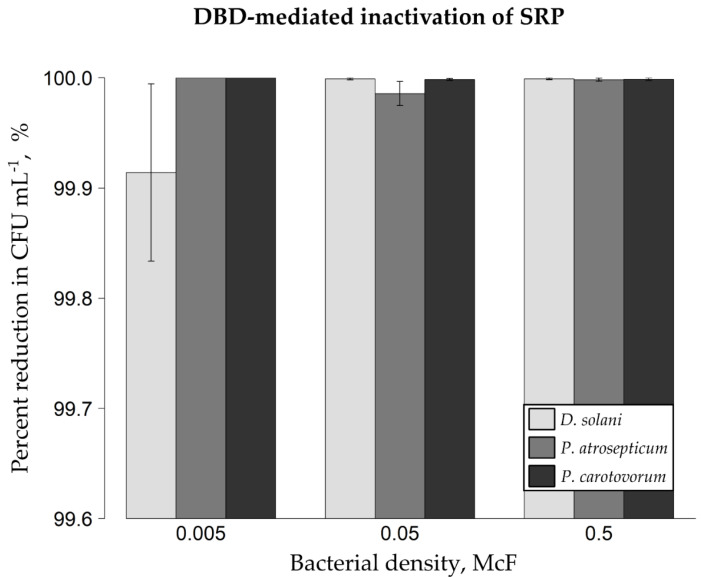
Inactivation efficacy of soft rot *Pectobacteriaceae* from mung bean seeds with the use of DBD as the NTAPP source. Percentage reductions in the amount of colony-forming units (CFU) per milliliter of Ringer buffer were shown. Either 0.5, 0.05 or 0.005 McF bacterial suspension had been used for inoculating disinfected mung bean seeds. Means ± standard errors are depicted. The experiment was performed in triplicate with three technical replicates each. No statistically significant differences between groups were observed (Kruskal-Wallis test followed by a post hoc analysis applying Fisher’s least significant difference criterion with Bonferroni correction at *p* < 0.05). *D. solani* IFB0099, *P. atrosepticum* IFB5103, and *P. carotovorum* IFB5118 strains were used. DBD—dielectric barrier discharge; NTAPP—non-thermal atmospheric pressure plasma; McF—McFarland scale; SRP—Soft rot *Pectobacteriaceae*.

**Figure 3 ijms-22-09256-f003:**
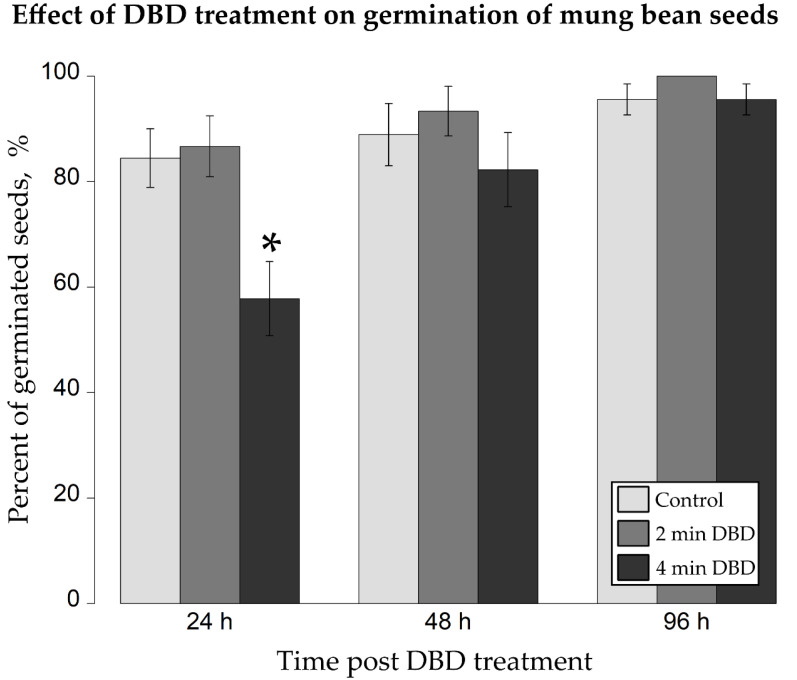
Effect of the NTAPP treatment on the germination rate of mung bean seeds. The seeds were exposed to DBD, being the NTAPP source, for 0, 2, or 4 min. The number of the germinated seeds was calculated after 24, 48, and 96 h of incubation. Bars correspond to the means ± standard errors. Asterisk shows a statistically significant difference in comparison to the untreated control sample (Student’s *t*-test at *p* < 0.05).

**Figure 4 ijms-22-09256-f004:**
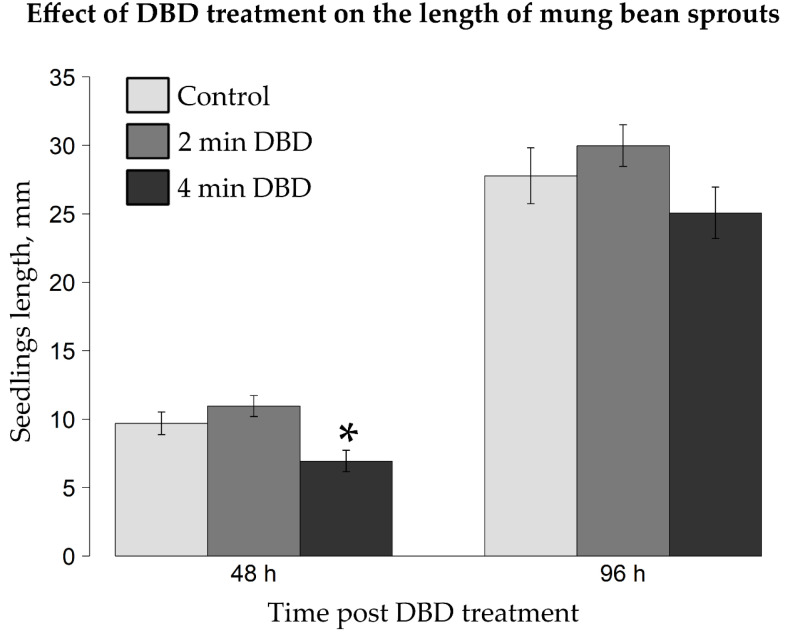
Effect of the NTAPP treatment on the length of mung bean sprouts. The seeds had been exposed to DBD, being the NTAPP source, for 0, 2, or 4 min. The resultant seedlings were measured after 48 h and 96 h of incubation. Bars depict means ± standard errors. Asterisk marks a statistically significant difference in comparison to the untreated control sample (Welch’s *t*-test at *p* < 0.05).

**Figure 5 ijms-22-09256-f005:**
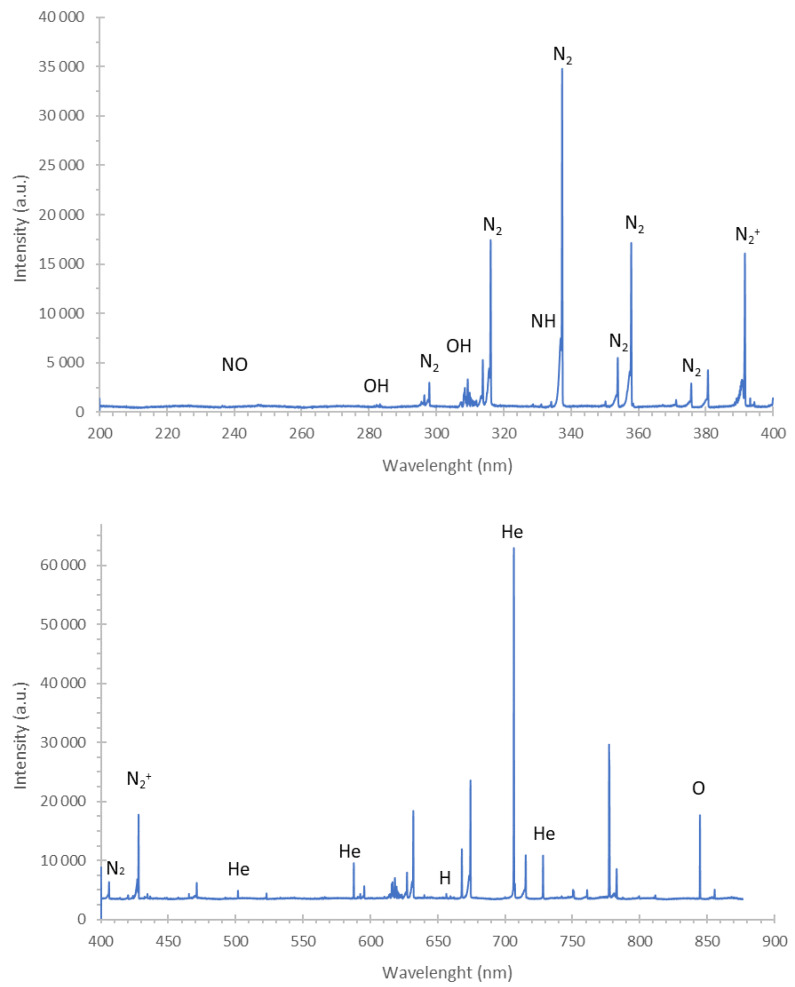
The emission spectra of DBD acquired in the range from 200 to 900 nm.

**Figure 6 ijms-22-09256-f006:**
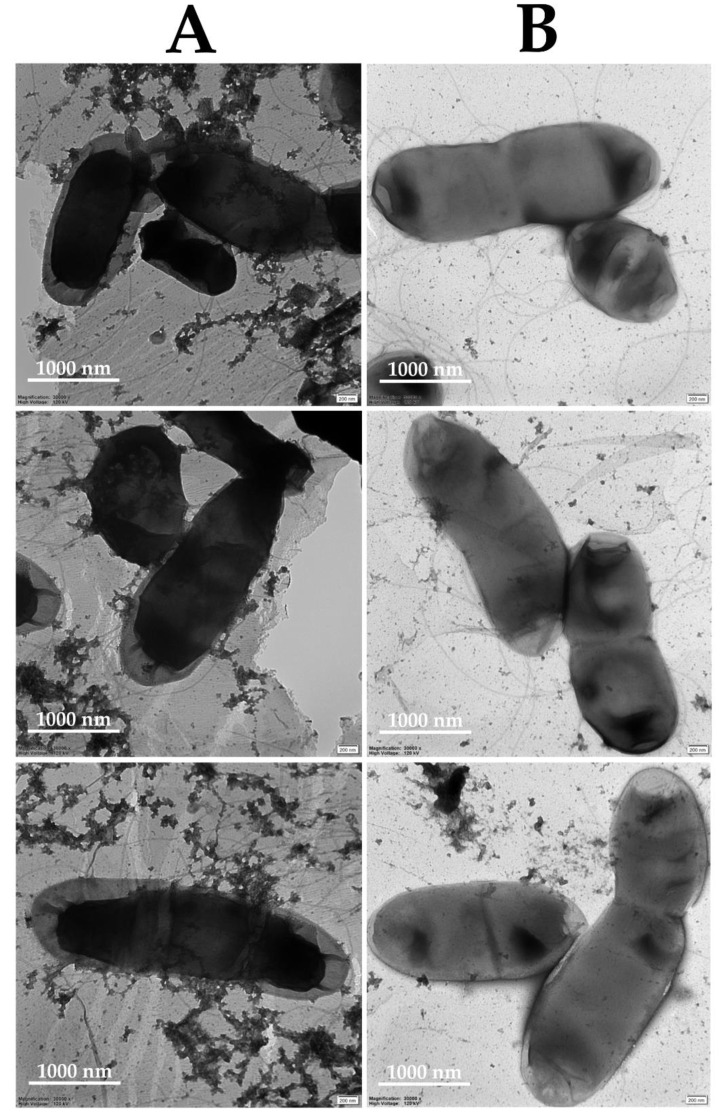
TEM photomicrographs of *D. solani* IFB0099 cells subjected to 5 min of the DBD treatment (**A**) and untreated controls (**B**). Magnification 30,000× at high voltage 120 kV.

**Figure 7 ijms-22-09256-f007:**
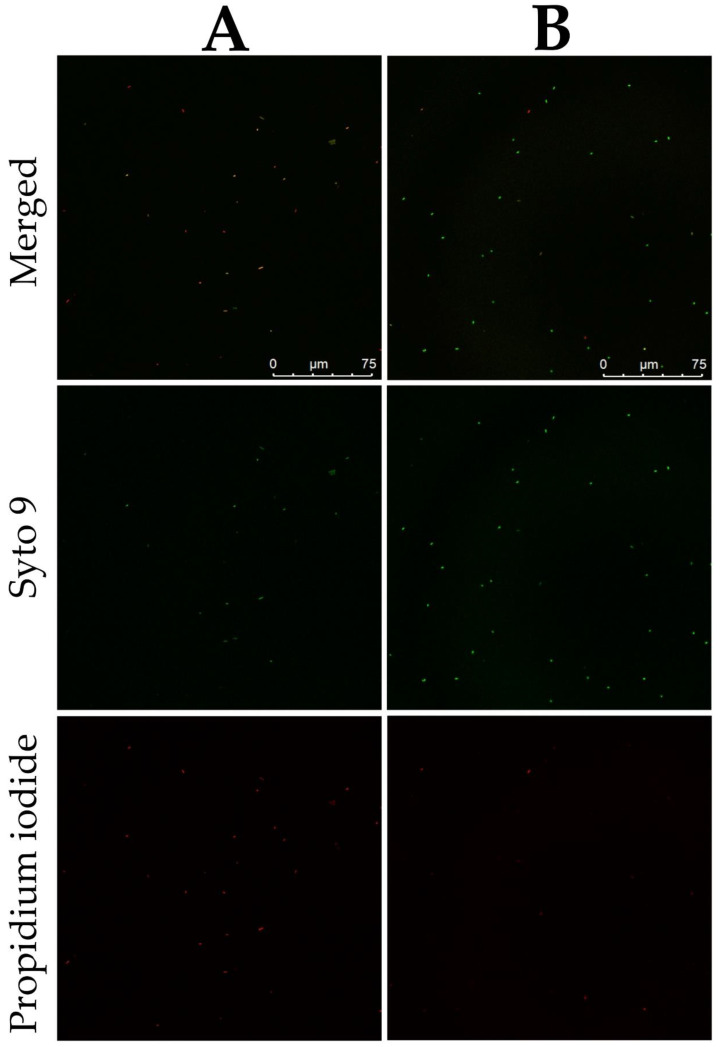
CLSM photomicrographs of *D. solani* IFB0099 cells subjected to 5-min action of NTAPP (**A**) and the untreated controls (**B**). The LIVE/DEAD BacLight Bacterial Viability Kit (Invitrogen, Waltham, MA, USA) was used for staining. *D. solani* IFB0099 cells, appearing red due to propidium iodide diffusion through the disrupted cellular membrane, are either dead or substantially damaged. *D. solani* IFB0099 cells, appearing green due to Syto 9 staining, remain viable. The imaging was performed with a CLSM Leica SP8X (Leica, Wetzlar, Germany) device with a 63× oil immersion lens.

**Figure 8 ijms-22-09256-f008:**
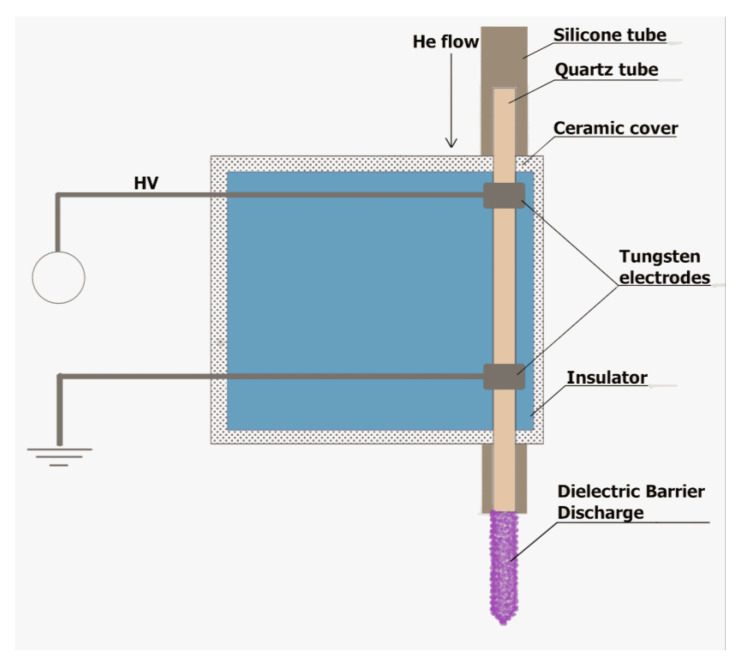
The scheme of the applied DBD-based portable plasma pencil.

**Table 1 ijms-22-09256-t001:** Plant pathogenic bacteria utilized in this study.

Plant Pathogenic Species	Strain Nos. ^a^	Disease	Country of Isolation	Year of Isolation	Reference
*Pectobacterium* *carotovorum*	IFB5118 SCRI136	Blackleg	USA	NA	SCRI collection [51]
*Pectobacterium* *atrosepticum*	IFB5103 SCRI1086	Blackleg and soft rot	Canada	1985	SCRI collection [51]
*Dickeya* *solani*	IFB0099 IPO2276 LMG28824	Blackleg and soft rot	Poland	2005	Slawiak et al. [52]

^a^ The presented strain numbers were attributed by the following international bacterial collections: IFB—bacterial collection of the Intercollegiate Faculty of Biotechnology of University of Gdansk and Medical University of Gdansk (Gdansk, Poland); SCRI—The James Hutton Institute bacterial collection (Dundee, Scotland); IPO—bacterial collection of the Institute for Phytopathological Research (Wageningen, The Netherlands); LMG—bacterial collection of the Laboratory of Microbiology in Gent (Gent, Belgium).

## Data Availability

The data presented in this study are available on a request from the corresponding author after gaining approval from the Technology Transfer Offices of University of Gdansk and Wroclaw University of Science and Technology. The data are not publicly available due to protection of intellectual property issues (see the Patents section).

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
