# Peer review of "Implementation of a Non-Thermal Atmospheric Pressure Plasma for Eradication of Plant Pathogens from a Surface of Economically Important Seeds"

_ijms, 2021, doi:10.3390/ijms22179256_

Round 1

Reviewer 1 Report

I believe the paper has an interesting approach both on the scientific and agronomic scope. I believe there is a considerable number of self-citations and it should be replaced with others, if possible. Overall, I think it is important to re-write and clarify the impact of non-thermal atmospheric pressure plasma on the plant pathogens, because I did not see it clearly that it will not have possible side effects to the plant and also to what extent is both the antimicrobial response of the pathogens and the impact of this method on products that will be entering the food chain.

Author Response

Responses to Reviewer 1

We are highly grateful for Reviewer’s 1 appreciation of our work and a kind comment ‘I believe the paper has an interesting approach both on the scientific and agronomic scope.’ Please find beneath our responses to the other suggestions:

  • I believe there is a considerable number of self-citations and it should be replaced with others, if possible.

R1) Thank you for this comment. There were 10 self-citations out of 56 references in the manuscript. According to the suggestion of the Reviewer 1 this number has been reduced by 4 in the resubmitted version of the manuscript, though some of the currently included self-citations are inevitable as either no other work was done previously is this exact field (Austin et al. 1988, Motyka et al. 2018, Dzimitrowicz et al. 2021), the references report previous methodologies or reaction discharge systems utilized also in this work (Terefinko et al. 2021)  or describe bacterial isolates used in the present study as well (Potrykus et al. 2014, Sławiak et al. 2009).

  • Overall, I think it is important to re-write and clarify the impact of non-thermal atmospheric pressure plasma on the plant pathogens, because I did not see it clearly that it will not have possible side effects to the plant and also to what extent is both the antimicrobial response of the pathogens and the impact of this method on products that will be entering the food chain.

R2) Thank you for this comment. The impact of NTAPPs on plant pathogens has been thoroughly described in the introduction section in terms of the current data on this matter (lines 96-118) and in the results&discussion section (lines 138-217). We agree that in terms of the effects of NTAPPs on plant material we solely focused on seeds in terms of seed germination efficacy and subsequent early growth of seedlings (Introduction: lines 103-105; 109-111; 114-118. Results&Discussion: lines 228-318), but it was due to the fact that we investigated the impact of NTAPP exposure of mung bean seeds and followed up seed germination rate and the resultant length of the seedlings. Thus, the effects of NTAPP treatment on physicochemical and organoleptic properties of plants, fruits or seeds designated for direct consumption were beyond the scope of our work. We, however, aim to follow up these aspects in our future studies. The topic related to physicochemical properties of the plasma-treated fresh products recalled by Reviewer 1 was only briefly mentioned in our work as this matter was investigated in the first study aiming at eradication of one of the plant pathogens that we also studied, namely P. carotovorum, from plant surfaces with the use of non-thermal plasma (Schnabel et al. 2015). We agree with the comment that this part was not clear and we have rewritten it in the resubmitted version of the manuscript (lines 212-220) accordingly to Reviewer’s suggestion.

Reviewer 2 Report

The authors reported the application of non-thermal atmospheric pressure plasma for the eradication of plant pathogens from the surface of seeds. The authors investigate a highly topical issue using a method of fighting pathogens that has been developed with good results in recent years. They reported very promising results.

The manuscript does not present particular problems. I suggest that the authors mitigate some of the statements in the abstract and along the manuscript. That is, the analyses carried out do not allow to state with absolute certainty that the non-thermal atmospheric pressure plasma “led to denaturation and aggregation of DNA, proteins, and ribosomes”. I suggest to add a: “probably”

Lines 284-286: if the statistical analysis does not support the differences between the theses, it is useless to point out that one thesis is slightly more efficient than another, they are all the same.

Author Response

Responses to Reviewer 2

We are highly grateful to Reviewer 2 for his kind comments on our work. Please find beneath our responses:

  • I suggest that the authors mitigate some of the statements in the abstract and along the manuscript. That is, the analyses carried out do not allow to state with absolute certainty that the non-thermal atmospheric pressure plasma “led to denaturation and aggregation of DNA, proteins, and ribosomes”. I suggest to add a: “probably”

R1) Thank you for this comment. According to the suggestion of Reviewer 2, the above-mentioned statements have been mitigated in the abstract and along the manuscript.

  • Lines 284-286: if the statistical analysis does not support the differences between the theses, it is useless to point out that one thesis is slightly more efficient than another, they are all the same.

R2) Thank you for this comment. The appropriate correction underlining lack of statistical significance for the reported numeric change has been made in the resubmitted version of the manuscript.